# Association of adenotonsillectomy with asthma and upper respiratory infection: A nationwide cohort study

**Jong-Yeup Kim**[1,2], **Inseok Ko**[2], **Ki Joon Park**[3], **Dong-Kyu Kim**[3,4]*

**1** Department of Otorhinolaryngology-Head and Neck Surgery, College of Medicine, Konyang University, Daejeon, Republic of Korea, **2** Department of Biomedical Informatics, College of Medicine, Konyang University, Daejeon, Republic of Korea, **3** Department of Otorhinolaryngology-Head and Neck Surgery, Division of Big Data and Artificial Intelligence, Chuncheon Sacred Heart Hospital, Hallym University College of Medicine, Chuncheon, Republic of Korea, **4** Institute of New Frontier Research, Hallym University College of Medicine, Chuncheon, Republic of Korea

* doctordk@naver.com

**Data Availability Statement:** The data underlying the results presented in the study are available from the national health claims database collected by the National Health Insurance Service (NHIS). The KNHIS-NSC dataset (NHIS-2018-2-143)

## Abstract

Adenotonsillectomy is a common paediatric surgery for treating obstructed breathing or recurrent inflammation; however, the long-term health consequences on the developing immune system are unknown. This study investigated the potential association between adenotonsillectomy and the development of asthma and upper respiratory infections (URI). This propensity score-matched retrospective cohort study utilized data from the National Sample Cohort 2002–2013. In the asthma cohort, we used a Cox-proportional hazards model to analyze the hazard ratio (HR) of adenotonsillectomy for asthma events. In the URI cohort, equivalence testing of postoperative visits for URI was performed. The margin of equivalence of the difference was set at -0.5–0.5. Asthma incidence was 66.97/1000 person-years in children who underwent adenotonsillectomy and 30.43/1000 person-years in those who did not. Adjusted asthma HRs were 2.25 (95% confidence interval, 1.96–2.57) in the adenotonsillectomy vs. non-adenotonsillectomy groups. In a subgroup analysis, children aged 5–9 years living in metropolitan areas showed a higher incidence of subsequent asthma than those of other ages and areas. However, any significant difference between the groups in terms of URI events in the 1–11-year postoperative period was not identified. Adenotonsillectomy in children is associated with an increased incidence of asthma, with no significant impact on postoperative visits for URI.

## Introduction

Every time we breathe and swallow, antigenic materials gain entry into our bodies. Although many inhaled or ingested antigens are harmless, some of these could contribute to the development of potentially dangerous conditions, requiring rapid and effective protective immune responses. Thus, from birth to adolescence, several mucosal immune systems develop in the upper and lower respiratory, gastrointestinal, and urogenital tracts [1,2]. Among these, the

comprises de-identified secondary data for research purposes, and can be requested from the NHIS website (nhiss.nhis.or.kr/bd/ay/bdaya001iv. do).

**Funding:** This research was supported by a grant of the Korea Health Technology R&D Project through the Korea Health Industry Development Institute (KHIDI), funded by the Ministry of Health & Welfare, Republic of Korea (grant number: HI17C2412). This research was also supported by the Bio & Medical Technology Development Program of the National Research Foundation (NRF), funded by the Korean government (MSIT) (NRF- 2017M3A9E8033231 to Dong-Kyu Kim). These funding sources had no role in the design of this study, and did not have any role during its execution, analyses, interpretation of the data, or decision to submit results.

**Competing interests:** The authors have declared that no competing interests exist.

nasopharyngeal-associated lymphoid tissues, which form the upper respiratory mucosal immune system, are arranged with a specific circular orientation around the wall of the throat called the Waldeyer's ring. These lymphoid structures comprise the adenoid (nasopharyngeal), tubal, palatine, and lingual tonsils. During childhood, these tissues play a major role in immunity as they are the first barrier of the host's resistance against pathogens [3–5].

Hypertrophy or frequent episodes of inflammation can occur within adenotonsillar tissues because they are continuously exposed to antigens, including many organisms and allergens that enter the body [6–8]. Thus, physicians often perform adenotonsillectomy to treat these pathologies and prevent or relieve the patient of the consequences, such as chronic rhinosinusitis, middle ear infection, and sleep apnoea [9–11]. The general risks of this surgery include those associated with the use of general anesthetic and those specific to the procedure, such as pain and immediate postoperative bleeding or as a result of a secondary infection within two weeks. However, contrasting evidence has been presented on the effect paediatric adenotonsillectomy has on the prevention of asthma and upper respiratory infections (URIs) [12–17]. Therefore, concerns have been raised regarding the need for an investigation into the long-term impact of adenoid and tonsil resections during childhood.

In this study, we investigated the association between pediatric adenotonsillectomy and the potential risk of asthma as well as the annual number of URIs in children (under 14 years). A nationwide representative sample of 1,025,340 subjects from the National Sample Cohort 2002–2013 of the Korea National Health Insurance Service (KNHIS-NSC) in South Korea was used. Since this nationwide population-based dataset contains information on the history of medical service utilization of more than 1 million Koreans, we were able to examine the association between pediatric adenotonsillectomy and the subsequent risks of specific comorbidities. We report that adenotonsillectomy during childhood increased the incidence of asthma, but we were unable to determine the risk it has upon the development of URIs.

## Materials and methods

### Patients and methods

This study adhered to the tenets of the Declaration of Helsinki and used the national health claims database collected by the National Health Insurance Service (NHIS). It was approved by the Institutional Review Board of Hallym Medical University, Chuncheon Sacred Hospital (No. 2019-02-005), and the need for written informed consent was waived as the KNHIS-NSC dataset (NHIS-2018-2-143) comprised de-identified secondary data for research purposes.

### Study population

In this study, we included children under 15 years of age. Children with adenotonsillectomy were identified as those who underwent an adenotonsillectomy (claim code: Q2280 or Q2281 [adenoidectomy] and Q2300 [tonsillectomy]) between January 2002 and December 2004. Among these children, those who underwent tonsillectomy or adenoidectomy alone were excluded. The control group, comprising 4 patients for every 1 patient who had undergone an adenotonsillectomy, was selected using propensity score-matching according to age, sex, residential area, and household income from January 2002 to December 2004. As a result, 648 children who underwent adenotonsillectomy and 2,592 controls were enrolled in the study cohort that evaluated the risk of asthma development; 1,039 children who underwent adenotonsillectomy and 4,156 controls were enrolled in the cohort that evaluated URI incidence. Flow chart for eligible study population was presented on Fig 1.

Each patient evaluated for the risk of asthma development was tracked until 2013, and the occurrences of asthma were obtained using the following International Statistical

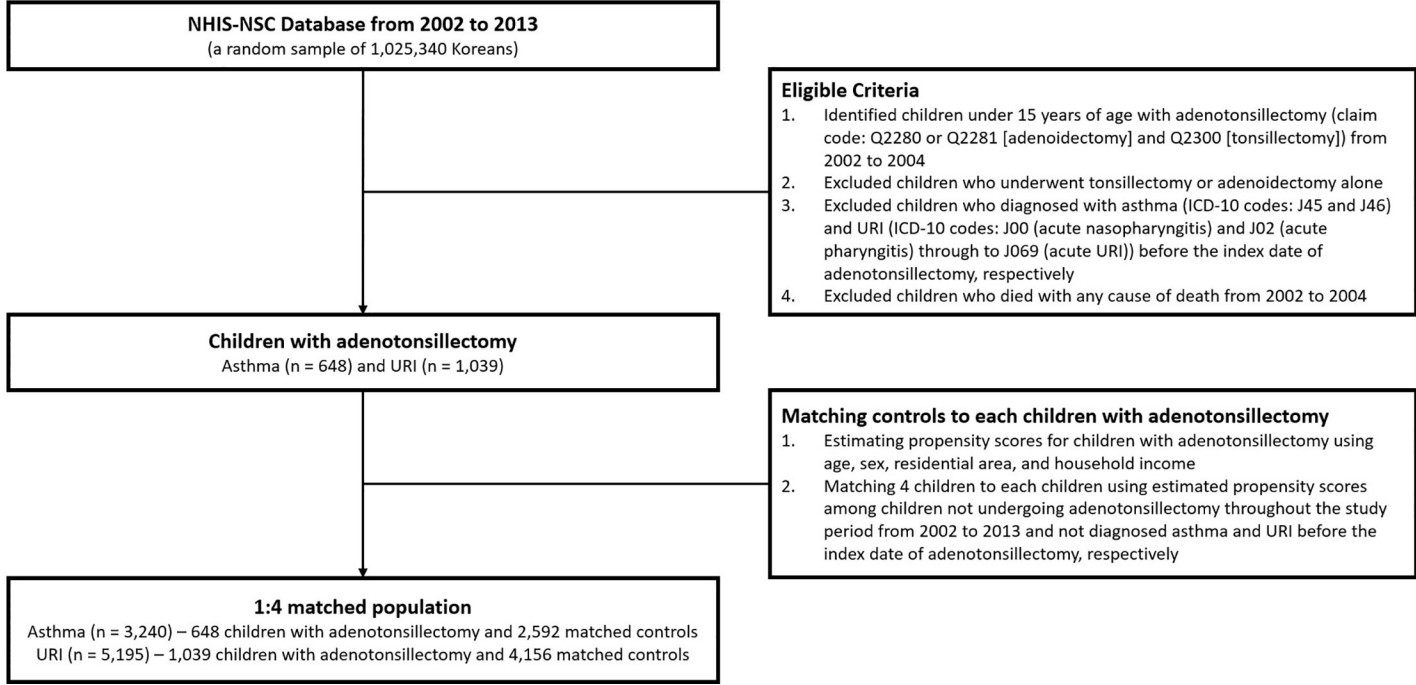

**Fig 1. The diagram presents the eligibility criteria for children who underwent adenotonsillectomy and their matched controls.**

Classification of Diseases and Related Health Problems (ICD)-10 codes: J45 and J46. Among these, the asthma group was defined by a physician-provided diagnosis on more than two occasions and the asthma-related medication history, including inhaled corticosteroids. Additionally, within the cohort evaluating URI incidence, URI was defined using the following ICD-10 codes: J00 (acute nasopharyngitis) and J02 (acute pharyngitis) through to J069 (acute URI). The number of visits to clinics or hospital for URIs was counted every year. Preoperative visits for URIs were counted for 2 years. The number of visits included in the history of the URI during the follow-up period was counted for each year (from post-operative year 1 to 11). Therefore, the participants who underwent adenotonsillectomy in 2002 were followed for 11 years, whereas those who underwent adenotonsillectomy in 2004 were followed for 9 years.

## Operational definition of endpoints and variables

The operational definitions of study endpoints were all-cause mortality or the incidence of asthma. All participants who had no events and were alive until December 31, 2013 were censored after this timepoint. The risks of asthma were compared between the adenotonsillectomy group and the control group using person-years at risk, which was defined as the duration between either the date of adenotonsillectomy or the same date for each patient matched with those in the adenotonsillectomy group (for the control group), and the patient's respective endpoint. Details of the patients' age, sex, residence, and household income were obtained from the database.

The study population was divided into 3 age groups (0–4, 5–9, and 10–14 years), 3 income groups (low [≤30.0% of the national median], middle [30.1%-69.9% of the national median], and high [≥70.0% of the national median]), and 3 residential areas (Seoul, the largest metropolitan region in South Korea, other metropolitan cities in South Korea, and small cities and rural areas).

## Statistical analysis

Descriptive and χ2 analyses were performed to identify sex, age groups, residence, and household income to evaluate the differences in the variables of the study participants. A 1:4 propensity score-matching was performed using the nearest neighbor matching method. Incidence rates per 1,000 person-years for asthma were obtained by dividing the number of patients with incidents of specific diseases by person-years at risk. The overall specific disease-free survival rate was determined using Kaplan–Meier survival curves with log-rank tests for the observation period. To identify whether adenotonsillectomy increased the risk of the occurrence of asthma, we used Cox proportional hazard regression analyses to calculate the hazard ratios (HRs) and 95% confidence intervals (CIs), adjusting for the other predictor variables.

Moreover, an equivalence test was used to compare the number of visits for URI (preoperative, and post-operative years 1 to 11) between the adenotonsillectomy and control groups. The null hypothesis was that visits for URI during the follow-up period would not be the same for both groups. In a previous meta-analysis, the pooled risk difference in URIs was -0.5 episodes per year [18]. Therefore, the margin of equivalence of difference (adenotonsillectomy–comparison) was set at -0.5–0.5 in this study. All statistical analyses were performed using R version 3.3.1 (R Foundation for Statistical Computing, Vienna, Austria), with a significance level of 0.05. Tableau Desktop Professional Edition (version 10.2) from the Tableau Software (Seattle, WA, USA) was also used to visualize patterns of the visits for URI based on the equivalence test results [19].

## Results

The patient characteristics for the two cohorts (for evaluation of asthma and URI incidence) have been presented in Table 1. The distributions of sex, age, residential area, and household income were similar in the adenotonsillectomy and control (non-adenotonsillectomy) groups,

**Table 1. Characteristics of the study subjects.**

| Variables | Asthma | | | | Upper respiratory infection | | | |
|---|---|---|---|---|---|---|---|---|
| | Comparison (n = 2,592) | AT (n = 648) | Effect size (95% CI) | χ2 | Comparison (n = 4,156) | AT (n = 1,039) | Effect size (95% CI) | χ2 |
| **Gender** | | | 0.000 (0.999–1.000) | 0.000 | | | 0.000 (0.999–1.000) | 0.000 |
| Male | 1,572 (60.6%) | 393 (60.6%) | | | 2,592 (62.4%) | 648 (62.4%) | | |
| Female | 1,020 (39.4%) | 255 (39.4%) | | | 1,564 (37.6%) | 391 (37.6%) | | |
| **Ages (years)** | | | 0.000 (0.999–1.000) | 0.000 | | | 0.002 (0.987–0.992) | 0.019 |
| 0–4 | 232 (9.0%) | 58 (9.0%) | | | 547 (13.2%) | 138 (13.3%) | | |
| 5–9 | 1,612 (62.2%) | 403 (62.2%) | | | 2,721 (65.5%) | 678 (65.3%) | | |
| 10–14 | 748 (28.9%) | 187 (28.9%) | | | 888 (21.4%) | 223 (21.5%) | | |
| **Residence** | | | 0.000 (0.999–1.000) | 0.000 | | | 0.002 (0.993–0.997) | 0.013 |
| Seoul | 556 (21.5%) | 139 (21.5%) | | | 909 (21.9%) | 228 (21.9%) | | |
| Other metropolitans | 652 (25.2%) | 163 (25.2%) | | | 987 (23.7%) | 248 (23.9%) | | |
| Rural and small cities | 1,384 (53.4%) | 346 (53.4%) | | | 2260 (54.4%) | 563 (54.2%) | | |
| **Household income** | | | 0.000 (0.999–1.000) | 0.000 | | | 0.002 (0.992–0.996) | 0.018 |
| ≤30.0 (low) | 260 (10.0%) | 65 (10.0%) | | | 466 (11.2%) | 115 (11.1%) | | |
| 30.1–69.9 (middle) | 992 (38.3%) | 248 (38.3%) | | | 1,591 (38.3%) | 398 (38.3%) | | |
| ≥70.0 (high) | 1,340 (51.7%) | 335 (51.7%) | | | 2,099 (50.5%) | 526 (50.6%) | | |

AT: Adenotonsillectomy

as these variables were used for sample matching; this indicated that the group matching in these cohorts had been performed appropriately.

## Effect of adenotonsillectomy on asthma

The cohort evaluated comprised 648 participants who had previously undergone adenotonsillectomy, and 2,592 controls (non-adenotonsillectomy). In this cohort, 23,692.6 person-years in the control group and 4,569.5 person-years in the adenotonsillectomy group were assessed for asthma events. Thus, the incidence of asthma in the adenotonsillectomy group was 66.97 per 1,000 person-years, which was higher than that of the control group at 30.43 per 1,000 person-years.

The results of univariate and multiple Cox regression models to analyse the HR for the development of asthma during the 11-year follow-up period have been presented in Table 2. As indicated by the results of the multiple Cox regression analysis of all variables, after adjusting for sociodemographic factors (sex, age, residential area, and household income), adenotonsillectomy was significantly associated with the prospective development of asthma (adjusted HR, 2.25; 95% CI, 1.96–2.57). The Kaplan–Meier survival curves indicate that children who underwent adenotonsillectomy developed asthma more frequently than those who did not (Fig 2).

Moreover, we conducted subgroup analyses for the development of asthma during the 11-year follow-up period according to the age groups and residence (Tables 3 and 4). After adjusting for other factors, we found that children aged 5–9 years (adjusted HR, 2.77; 95% CI, 2.35–3.27) showed a higher incidence of asthma than those aged 10–14 years. Additionally, children who lived in Seoul, the largest metropolitan region considered, showed a higher incidence of the development of asthma (adjusted HR, 2.58; 95% CI, 1.95–3.41) than those who lived in other areas.

**Table 2. Incidence per 1,000 person-years and HR (95% CIs) of asthma during the 11-year follow-up period.**

| Variables | N | Case | Person-years | Incidence | Unadjusted hazard ratio (95% confidence intervals) | Adjusted hazard ratio (95% confidence intervals) |
|---|---|---|---|---|---|---|
| **Group** | | | | | | |
| Comparison | 2,592 | 721 | 23,692.6 | 30.43 | 1 (ref) | 1 (ref) |
| Adenotonsillectomy | 648 | 306 | 4,569.5 | 66.97 | 2.19 (1.91–2.5) | 2.25 (1.96–2.57) |
| **Gender** | | | | | | |
| Male | 1,965 | 635 | 16,995.6 | 37.36 | 1 (ref) | 1 (ref) |
| Female | 1,275 | 392 | 11,266.5 | 34.79 | 0.93 (0.82–1.06) | 0.96 (0.84–1.09) |
| **Ages (years)** | | | | | | |
| 0–4 | 290 | 180 | 1,767.0 | 101.87 | 1 (ref) | 1 (ref) |
| 5–9 | 2,015 | 637 | 17,728.8 | 35.93 | 0.36 (0.31–0.2) | 0.36 (0.31–0.43) |
| 10–14 | 935 | 210 | 8,766.3 | 23.96 | 0.24(0.31–0.2) | 0.36 (0.31–0.43) |
| **Residence** | | | | | | |
| Seoul | 695 | 236 | 5,985.0 | 39.43 | 1 (ref) | 1 (ref) |
| Other metropolitans | 815 | 254 | 7,166.7 | 35.44 | 0.9 (0.75–1.07) | 0.23 (0.19–0.28) |
| Rural and small cities | 1,730 | 537 | 15,110.4 | 35.54 | 0.9 (0.77–1.05) | 0.93 (0.78–1.11) |
| **Household income** | | | | | | |
| ≤30.0 (low) | 325 | 116 | 2,803.9 | 41.37 | 1 (ref) | 1 (ref) |
| 30.1–69.9 (middle) | 1,240 | 386 | 10,958.6 | 35.22 | 0.85 (0.69–1.05) | 0.93 (0.8–1.08) |
| ≥70.0 (high) | 1,675 | 525 | 14,499.6 | 36.21 | 0.88 (0.72–1.07) | 0.87 (0.71–1.07) |

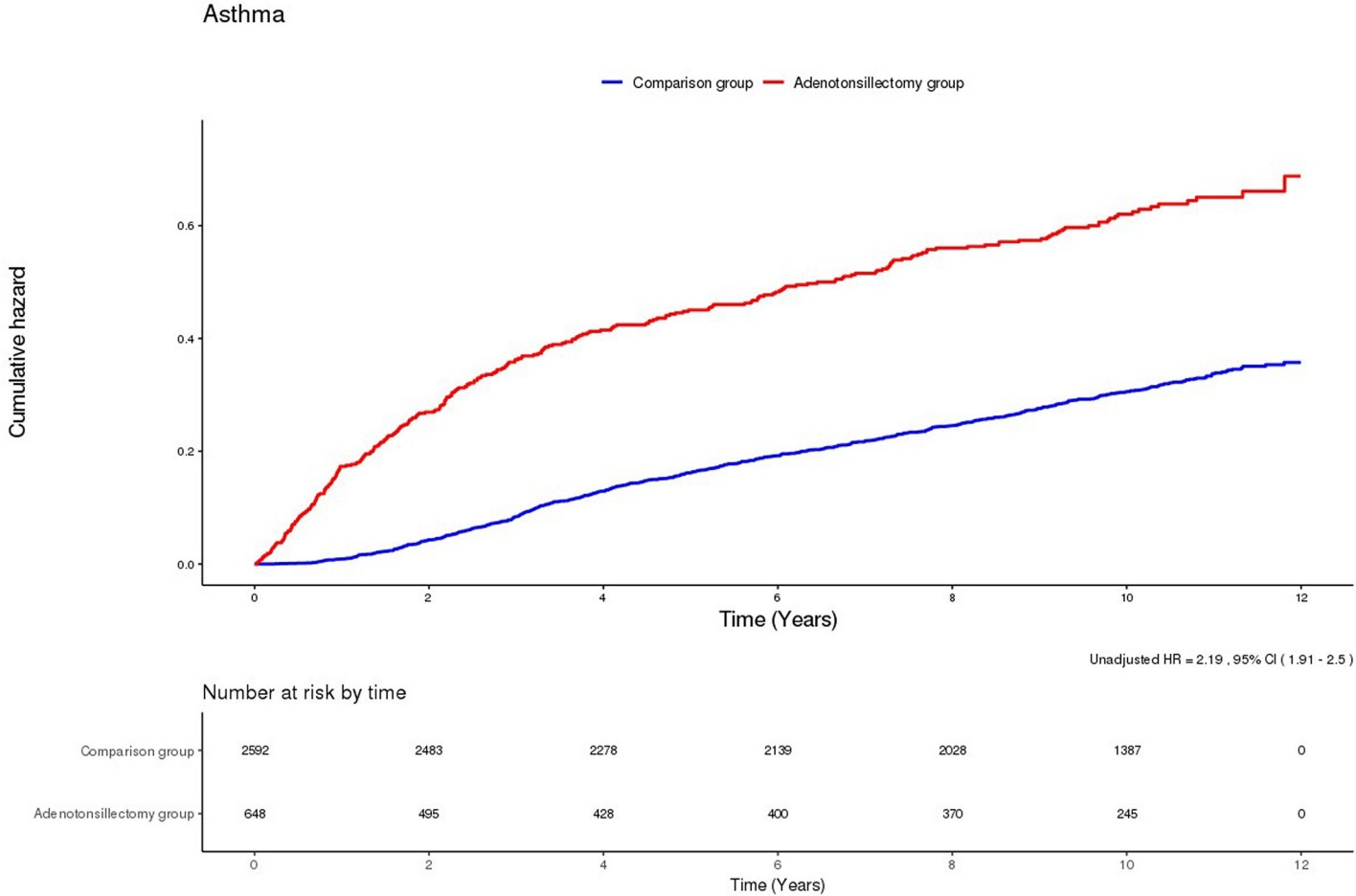

**Fig 2. Kaplan-Meier survival curves and log-rank tests for the development of asthma.**

## Effect of adenotonsillectomy on URI

The cohort evaluated in this case consisted of 1,039 patients having undergone adenotonsillectomy and 4,156 controls (non-adenotonsillectomy). In addition to similar distributions of sex, age, residential area, and household income, the number of visits for preoperative URIs was also identical in both groups (Table 5). When we compared the visits for URIs during the follow-up period, no difference was observed between the adenotonsillectomy and control groups from post-operative year 1 to 11 (-0.5 < 95% CI of difference < 0.5). We found that visits regarding URIs gradually decreased from the pre- to post-operative years in the adenotonsillectomy group. However, a similar change was also observed in the comparison group.

**Table 3. Hazard ratios of asthma by age.**

| Age (years) | <4 | | 5–9 | | 10–14 | |
|---|---|---|---|---|---|---|
| | **Comparison** | **AT** | **Comparison** | **AT** | **Comparison** | **AT** |
| *Unadjusted hazard ratio (95% confidence intervals)* | 1 (ref) | 1.53 (1.07–2.19) | 1 (ref) | 2.78 (2.35–3.27) | 1 (ref) | 1.59 (1.17–2.17) |
| *Adjusted hazard ratio (95% confidence intervals)* | 1 (ref) | 1.52 (1.07–2.18) | 1 (ref) | 2.77 (2.35–3.27) | 1 (ref) | 1.60 (1.18–2.17) |

AT: Adenotonsillectomy

**Table 4. Hazard ratios of asthma by residence.**

| Residence | Seoul | | Other areas | |
|---|---|---|---|---|
| | Comparison | Adenotonsillectomy | Comparison | Adenotonsillectomy |
| *Unadjusted hazard ratio* (95% confidence intervals) | 1 (ref) | 2.48 (1.88–3.28) | 1 (ref) | 2.11 (1.81–2.46) |
| *Adjusted hazard ratio* (95% confidence intervals) | 1 (ref) | 2.58 (1.95–3.41) | 1 (ref) | 2.16 (1.85–2.52) |

Furthermore, in the subgroup analysis according to age or residence, we found no significant difference in the number of postoperative visits for URI between the adenotonsillectomy and control groups in all age or residence categories (patients <4, 5–9, and 10–14 years old; Seoul and other areas) (Fig 3A and 3B; S1–S5 Tables).

## Discussion

Adenotonsillectomy is one of the most common surgical procedures performed in children and its indications are still controversial. However, adenotonsillectomy is generally accepted to be indicated in tonsil and adenoid tissues for causing: 1) obstructive sleep apnoea, 2) refractory or recurrent sinusitis or middle ear infections, and 3) recurrent infection of the tonsils and/or adenoids. To the best of our knowledge, the present study is the first study based on data from a nationwide representative cohort to evaluate the risk of asthma and the frequency of URI after adenotonsillectomy in children. We observed an association between adenotonsillectomy and an increased incidence of asthma. Additionally, we compared postoperative visits for URIs between children who underwent adenotonsillectomy and those who did not, however we note no association.

### Implications for asthma development and control

Asthma is a common inflammatory disease of the lower airways [20]. Respiratory infection and/or allergen exposure are frequently indicated as triggers for the exacerbation of asthma in children. Thus, a number of observational studies have demonstrated a positive clinical effect of adenotonsillectomy on paediatric asthma control [21–23]. These studies suggest that removing the adenoid and tonsils may reduce the effect of stressors on the lower airway, leading to decreased inflammation and improved asthma control. However, although

**Table 5. Comparison of equivalence test for upper respiratory infections in the pre- and post-operative periods.**

| Variable | Comparison (mean ± SD) | Adenotonsillectomy (mean ± SD) | 95% CI of the difference (0.5) | P value |
|---|---|---|---|---|
| Pre-op visit | 5.0 ± 5.1 | 5.0 ± 5.2 | -0.35 to 0.37 | 0.954 |
| Post-op 1 y visit | 2.6 ± 2.5 | 2.9 ± 2.5 | 0.05 to 0.39 | 0.012 |
| Post-op 2 y visit | 2.3 ± 2.4 | 2.5 ± 2.4 | 0.04 to 0.36 | 0.015 |
| Post-op 3 y visit | 2.0 ± 2.2 | 2.1 ± 2.2 | -0.02 to 0.28 | 0.078 |
| Post-op 4 y visit | 1.8 ± 2.1 | 1.9 ± 2.1 | -0.05 to 0.24 | 0.185 |
| Post-op 5 y visit | 1.8 ± 2.1 | 2.0 ± 2.2 | 0.07 to 0.36 | 0.004 |
| Post-op 6 y visit | 1.7 ± 2.1 | 1.8 ± 2.0 | -0.03 to 0.24 | 0.136 |
| Post-op 7 y visit | 1.5 ± 1.9 | 1.7 ± 2.0 | 0.09 to 0.35 | 0.001 |
| Post-op 8 y visit | 1.3 ± 1.8 | 1.5 ± 1.9 | 0.04 to 0.29 | 0.012 |
| Post-op 9 y visit | 1.2 ± 1.6 | 1.3 ± 1.8 | -0.02 to 0.22 | 0.089 |
| Post-op 10 y visit | 0.9 ± 1.8 | 1.1 ± 2.1 | -0.01 to 0.19 | 0.065 |
| Post-op 11 y visit | 0.2 ± 0.9 | 0.3 ± 1.0 | -0.03 to 0.11 | 0.283 |

Op: operation, SD: Standard deviation, Difference: adenotonsillectomy group—comparison group, CI: Confidence interval

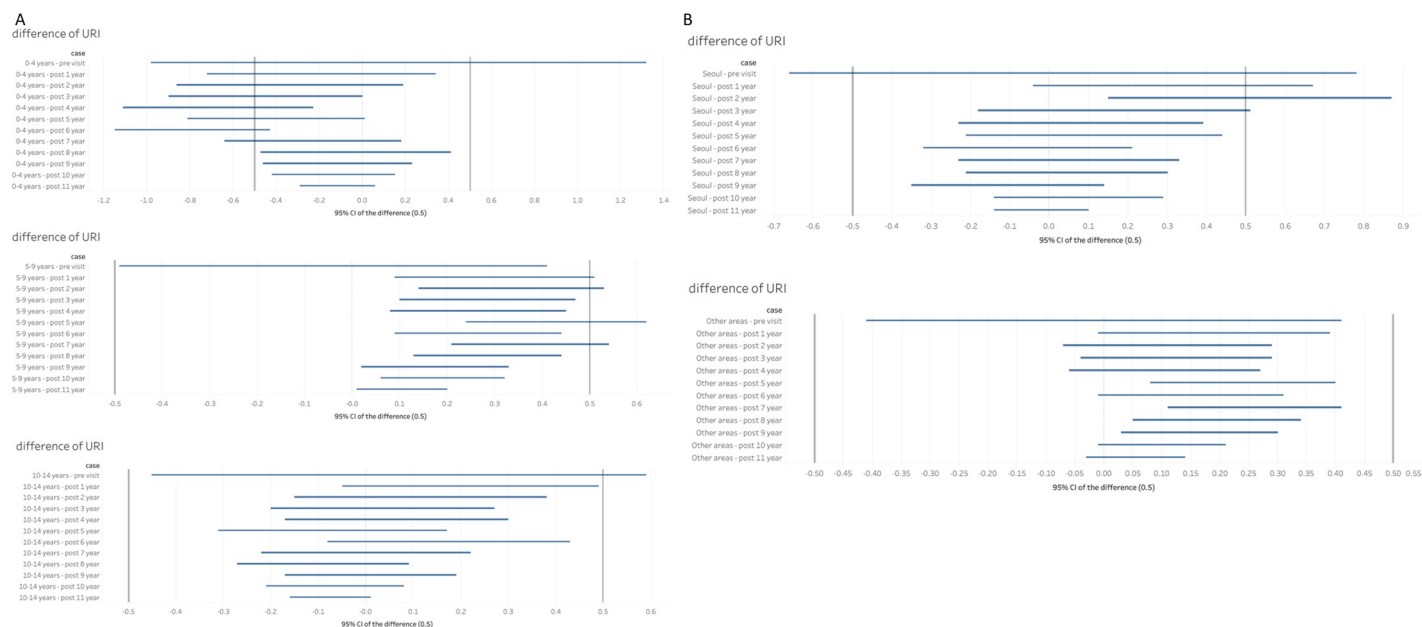

**Fig 3.** Differences in the mean values for pre- and post-operative upper respiratory infection according to (A) age or (B) residence.

adenotonsillectomy has clear effects on the status of asthma control, the effect of this procedure on the subsequent development of asthma has not yet been fully elucidated. To date, few studies have reported on the association between adenoidectomy and asthma development. One cohort study previously demonstrated that early-life adenoidectomy may contribute to the subsequent development of asthma in children [12], whereas another randomized control study revealed that adenoidectomy did not promote the occurrence of asthma or allergy [13]. However, these studies have noteworthy limitations. For example, the latter had a relatively short follow-up period and the enrolled patients were not representative of the whole population because the study was conducted in a single center. Moreover, these studies investigated the subsequent risk of asthma following adenoidectomy, not adenotonsillectomy, and thus has excluded any effect the removal of the tonsils may have. The tonsil is located at the entrance to the respiratory system, and functions as part of the mucosal-associated lymphoid tissue by providing lymphocytes to identify and challenge pathogenic organisms. For example, these immature B lymphocytes approach foreign antigens and undergo the initial stages of differentiation towards antibody-producing plasma cells. Thus, adenotonsillectomy may be more influential on early-life immune function than adenoidectomy, as adenotonsillectomy usually involves the surgical removal of the whole palatine and nasopharyngeal tonsils from their investing tissues.

In the present study, we found that adenotonsillectomy was significantly associated with the prospective development of asthma, as indicated by the results of multiple Cox regression analyses of all variables. Specifically, we observed that children aged 5–9 years showed a higher incidence of the development of asthma than those aged 10–14 years. Although the adjusted HR in children aged 4 years and below is lower than that in children aged 5–9 years, we thought this finding should be interpreted with caution, because the sample number of children enrolled below 4 years of age was inadequate. Concurrent with our findings, one recent population-based cohort study revealed that the relative risk of asthma was 1.45 in children following adenoidectomy compared with controls [14]. Since the tonsils and adenoids are part of

the lymphatic system and play key roles both in the normal development of the immune system and in pathogen screening during childhood and early life [9], it is not unexpected that their removal may be associated with altered immune profiles. Consistent with our findings, numerous reports indicate that altering early life immune pathways may have lasting effects on adult health [24–28]. In addition, we observed that the adjusted HRs of asthma were significantly higher in children who had undergone adenotonsillectomy and in those who lived in the largest metropolitan region of Seoul, than in those who lived in other regions. This finding is also supported by the immune hygiene hypothesis [29].

## Implications for URIs

The palatine and nasopharyngeal tonsils are considered to play an important role in the causation of chronic or recurrent acute throat infections. The hypothesis that children with no palatine and nasopharyngeal tonsils would experience a reduced number and/or severity of future throat infections therefore remains. However, many clinicians often observe that some children who undergo adenotonsillectomy continue to suffer from pharyngitis and sore throats without any sign of tonsil and/or adenoid tissue regrowth. Therefore, to date, it is unclear whether the removal of tonsils and adenoid tissues has a preventive effect on the incidence of URI. Moreover, numerous studies have showed a lack of a preventive effect of adenoidectomy or tonsillectomy on the number of postoperative visits for URIs [14,15,30,31]. Similar to those studies, we found no difference between the adenotonsillectomy and control groups during the 1- to 11-year post-operative follow-up visits.

## Study strengths and limitations

The present study, which was based on data collected from the KNHIS-NSC, which it has been previously confirmed as enables us to effectively analyze specific disease incidents in South Korea [32]. First, prior studies on the association between adenoidectomy or tonsillectomy and its comorbidities were cross-sectional or cohort studies with a relatively short observation period. However, our study reveals an association between the subsequent development of asthma and URI in children who underwent adenotonsillectomy using long-term, longitudinal nation-wide cohort datasets. Second, to control confounding factors, we included only children who underwent adenotonsillectomy, and excluded those who did not. Finally, we also analyzed the effect of adenotonsillectomy on its potential comorbidities, such as asthma and URI according to the age, gender, and residence area of the participants.

This study had some limitations. First, we had no access to data regarding asthma severity, such as that determined by a pulmonary function test or an asthma-related questionnaire, and that of the URIs, such as the presence of serologic inflammation markers or medical records. Therefore, we could not perform subgroup analysis regarding the severity. Second, the accuracy of the diagnosis of paediatric asthma has some fundamental issues, as follows: (1) diagnostic pulmonary function testing including methacholine or exercise bronchial provocation testing is difficult to perform in children, and (2) virally induced wheezing during early childhood could easily be misdiagnosed as asthma. Thus, to overcome this issue, we applied the operational definitions of pediatric asthma, as described in the Methods section. Third, we were unable to access other specific health data, such as body mass index, lipid profiles, and the exposure to second-hand smoke at home. Data such as these could contribute to the incidence of asthma and URI. Therefore, these possible confounding factors could not be controlled in this study. Finally, because this was a retrospective cohort study, we could not directly examine and analyse the mechanisms underlying the association between adenotonsillectomy and its comorbidities. Future clinical studies investigating a wider range of factors,

diagnostic criteria, and objective disease severity can provide additional evidence for the link between adenotonsillectomy and its comorbidities.

## Conclusions

In conclusion, we investigated a possible link between paediatric adenotonsillectomy and the prospective development of asthma as well as compared the postoperative visits for URIs between the adenotonsillectomy and non-adenotonsillectomy groups. We observed that children who underwent adenotonsillectomy had a higher risk of developing asthma during an 11-year follow-up period, whereas this surgical procedure had no benefit in preventing URIs. Although we observed a decrease in visits regarding URIs during the 11-year follow-up period via data from insurance claims, this trend was also observed in the control group, which showed a similar scale. These findings suggest that clinicians treating children with adenotonsillectomy may need to pay careful attention as it may infer an increased risk of developing asthma. However, it should be recognized that the incidence of URI remains unchanged after adenotonsillectomy.

## Supporting information

**S1 File. Online repository material.**
(DOCX)

**S1 Table. Equivalence tests for upper respiratory infections in the postoperative period in patients under 4 years.**
(DOCX)

**S2 Table. Equivalence tests for upper respiratory infections in the postoperative period in patients aged 5–9 years.**
(DOCX)

**S3 Table. Equivalence tests for upper respiratory infections in the postoperative period in patients aged 10–14 years.**
(DOCX)

**S4 Table. Equivalence tests for upper respiratory infections in the postoperative period in patients living in Seoul.**
(DOCX)

**S5 Table. Equivalence tests for upper respiratory infections in the postoperative period in patients living in other areas.**
(DOCX)

## Author Contributions

**Conceptualization:** Jong-Yeup Kim, Dong-Kyu Kim.

**Data curation:** Dong-Kyu Kim.

**Formal analysis:** Inseok Ko.

**Funding acquisition:** Jong-Yeup Kim, Dong-Kyu Kim.

**Investigation:** Jong-Yeup Kim.

**Methodology:** Inseok Ko.

**Project administration:** Jong-Yeup Kim, Dong-Kyu Kim.

**Resources:** Jong-Yeup Kim.

**Software:** Inseok Ko.

**Supervision:** Dong-Kyu Kim.

**Validation:** Jong-Yeup Kim.

**Visualization:** Inseok Ko.

**Writing – original draft:** Dong-Kyu Kim.

**Writing – review & editing:** Ki Joon Park, Dong-Kyu Kim.

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
