## [Decision Letter · Decision Letter 0]

26 Jun 2020

PONE-D-20-16108

Association of adenotonsillectomy with asthma and upper respiratory infection: A Nationwide Cohort Study

PLOS ONE

Dear Dr. Kim,

Thank you for submitting your manuscript to PLOS ONE. After careful consideration, we feel that it has merit but does not fully meet PLOS ONE’s publication criteria as it currently stands. Therefore, we invite you to submit a revised version of the manuscript that addresses the points raised during the review process.

We look forward to receiving your revised manuscript.

Kind regards,

Giovanni Cammaroto

Academic Editor

PLOS ONE

Journal Requirements:

This research was supported by the Korea Health Technology R&D Project grant through the Korea Health Industry Development Institute (KHIDI), funded by the Ministry of Health & Welfare (H17C2412 to Jong-Yeup Kim). This research was also supported by the Bio & Medical Technology Development Program of the National Research Foundation (NRF), funded by the Korean government (MSIT) (NRF- 2017M3A9E8033231 to Dong-Kyu Kim). These funding sources had no role in the design of this study, and did not have any role during its execution, analyses, interpretation of the data, or decision to submit results.

"The funders had no role in study design, data collection and analysis, decision to publish, or preparation of the manuscript"

Reviewers' comments:

Reviewer's Responses to Questions

**Comments to the Author**

1. Is the manuscript technically sound, and do the data support the conclusions?

Reviewer #1: Yes

Reviewer #2: Yes

2. Has the statistical analysis been performed appropriately and rigorously? 

Reviewer #1: Yes

Reviewer #2: Yes

3. Have the authors made all data underlying the findings in their manuscript fully available?

Reviewer #1: Yes

Reviewer #2: No

4. Is the manuscript presented in an intelligible fashion and written in standard English?

Reviewer #1: Yes

Reviewer #2: Yes

5. Review Comments to the Author

Reviewer #1: This interesting work is based on data from a nationwide representative cohort to evaluate the risk of asthma and the frequency of URI after adenotonsillectomy in children.

The paper is worth publication following minor changes, especially better highlighting the limits of the study stated below.

The group matching in these cohorts had been performed appropriately. This work analyzes the association on the national territory dividing the sample by age, income and territories of residence. Patient follow-up time was appropriate to evaluate the risk of asthma and the frequency of URI after adenotonsillectomy in children. The data concerning the risk of incidence of asthma in patients undergoing adenotonsillectomy does not specify the degree of severity, owing to lack of access to asthma data. Unfortunately, the gradual decrease in adherence to visits for the evaluation of the URI in the post-operative compared to the pre-operative, in both groups, has resulted in a loss of interesting data. However, as highlighted in the work, some finding should be interpreted with caution, because the sample number of children enrolled below 4 years of age was inadequate. Due to the inaccessibility to other specific health data, such as body mass index, lipid profiles, and the exposure to second-hand smoke at home, these possible confounding factors could not be controlled. Furthermore, it should be considered that the accuracy of the diagnosis of pediatric asthma has some fundamental issues. Overall, the study is interesting and confirms what is already known in the literature between the association of adenotonsillectomy in children and the incidence of asthma. The analysis of the subgroups has shown a higher incidence of asthma in patients between 5 and 9 years in metropolitan areas, this result is very interesting and is supported by immune hygiene hypothesis.

Reviewer #2: 1. OVERALL: This article is an original research regarding the Association of adenotonsillectomy with asthma and upper respiratory infection (URI). Using data from the National Sample Cohort 2002-2013 of the Korean National Health Insurance Service (KNHIS-NSC), the authors showed that Adenotonsillectomy in children is associated with an increased incidence of asthma in the years that follows the surgery without a significant impact on post-operative visits for URI. In our opinion the topic of this paper is sound and the study’s design satisfies PLOS ONE’s criteria for publication.

2. ABSTRACT: In line 31 the authors could highlight that the propensity score-matched cohort is retrospective.

3. INTRODUCTION: Satisfactory.

4. MATERIALS AND METHODS: In this section it would be appropriate to cite a source containing more specific information about the database KNHIS-NSC (e.g. who, how and for what purpose it was made, is it currently updated?). We also suggest to summarize it in a visual diagram and put emphasis on how the population was selected.

5. RESULTS: Satisfactory, the data collected are clear and well presented.

6. DISCUSSION: The authors rightly underline that indications for adenotonsillectomy are still controversial and in line 205-209 this is summarized in 3 different points. In our opinion, a good general classification separates indications for adenotonsillectomy in 2 categories: obstruction and infection (see article of Paradise et al. 2020 https://www.uptodate.com/contents/tonsillectomy-and-or-adenoidectomy-in-children-indications-and-contraindications). However, in line 208 the authors state that 3 different infection events of the tonsil in a single year, despite adequate medical treatment, are sufficient to undertake surgery: this number is objectively low in absence of specific

comorbidities or complications. In line 278, citation of articles 32-35 is irrelevant to the topic of this research.

7. CONCLUSIONS: Conclusions are in line with the results.

8. REFERENCES: Citations 32-35 are irrelevant to the topic of this research.

9. TABLES: Satisfactory

10. FIGURES: Figure 2 is not clear: an higher quality image is recommended.

The authors of this peer-review declares that they have no conflict of interest.

6. PLOS authors have the option to publish the peer review history of their article (what does this mean?). If published, this will include your full peer review and any attached files.

Reviewer #1: No

Reviewer #2: No

---

## [Author Response · Author response to Decision Letter 0]

7 Jul 2020

Responses to the 1st Reviewer’s Comments

My coauthors and I greatly appreciate the reviewer’s comments and we respond to them as below. We highlighted sentences that we changed or added.

This interesting work is based on data from a nationwide representative cohort to evaluate the risk of asthma and the frequency of URI after adenotonsillectomy in children. The paper is worth publication following minor changes, especially better highlighting the limits of the study stated below. The group matching in these cohorts had been performed appropriately. This work analyzes the association on the national territory dividing the sample by age, income and territories of residence. Patient follow-up time was appropriate to evaluate the risk of asthma and the frequency of URI after adenotonsillectomy in children. The data concerning the risk of incidence of asthma in patients undergoing adenotonsillectomy does not specify the degree of severity, owing to lack of access to asthma data. Unfortunately, the gradual decrease in adherence to visits for the evaluation of the URI in the post-operative compared to the pre-operative, in both groups, has resulted in a loss of interesting data. However, as highlighted in the work, some finding should be interpreted with caution, because the sample number of children enrolled below 4 years of age was inadequate. Due to the inaccessibility to other specific health data, such as body mass index, lipid profiles, and the exposure to second-hand smoke at home, these possible confounding factors could not be controlled. Furthermore, it should be considered that the accuracy of the diagnosis of pediatric asthma has some fundamental issues. Overall, the study is interesting and confirms what is already known in the literature between the association of adenotonsillectomy in children and the incidence of asthma. The analysis of the subgroups has shown a higher incidence of asthma in patients between 5 and 9 years in metropolitan areas, this result is very interesting and is supported by immune hygiene hypothesis.

Answer: Thank you for your kind and favorable review. Again, we sincerely appreciate the evaluation of the referees. 

Responses to the 2nd Reviewer’s Comments

My coauthors and I greatly appreciate the reviewer’s questions and comments and we respond to them as below. We highlighted sentences that we changed or added.

The authors present a generally well done and thoughtful study performed in a very powerful dataset. However, there are several areas in need of clarification from a statistical point of view as detailed below.

1. OVERALL: This article is an original research regarding the Association of adenotonsillectomy with asthma and upper respiratory infection (URI). Using data from the National Sample Cohort 2002-2013 of the Korean National Health Insurance Service (KNHIS-NSC), the authors showed that Adenotonsillectomy in children is associated with an increased incidence of asthma in the years that follows the surgery without a significant impact on post-operative visits for URI. In our opinion the topic of this paper is sound and the study’s design satisfies PLOS ONE’s criteria for publication.

Answer: Thank you for your kind and favorable review.

2. ABSTRACT: In line 31 the authors could highlight that the propensity score-matched cohort is retrospective.

Answer: As you commented, we modified this sentence as follows: “This propensity score-matched retrospective cohort study”. 

3. INTRODUCTION: Satisfactory.

Answer: Thank you for your comment.

4. MATERIALS AND METHODS: In this section it would be appropriate to cite a source containing more specific information about the database KNHIS-NSC (e.g. who, how and for what purpose it was made, is it currently updated?). We also suggest to summarize it in a visual diagram and put emphasis on how the population was selected.

Answer: As you recommended, we added the reference regarding the database KNHIS-NSC and the flow chart as a modified figure 1.

5. RESULTS: Satisfactory, the data collected are clear and well presented.

6. DISCUSSION: The authors rightly underline that indications for adenotonsillectomy are still controversial and in line 205-209 this is summarized in 3 different points. In our opinion, a good general classification separates indications for adenotonsillectomy in 2 categories: obstruction and infection (see article of Paradise et al. 2020 https://www.uptodate.com/contents/tonsillectomy-and-or-adenoidectomy-in-children-indications-and-contraindications). However, in line 208 the authors state that 3 different infection events of the tonsil in a single year, despite adequate medical treatment, are sufficient to undertake surgery: this number is objectively low in absence of specific comorbidities or complications. In line 278, citation of articles 32-35 is irrelevant to the topic of this research.

Answer: As you commented, we modified this sentence as follows:“3) recurrent infection of the tonsils and/or adenoids”. Additionally, we remove these references (32-35).

7. CONCLUSIONS: Conclusions are in line with the results.

Answer: Thank you for your comment.

8. REFERENCES: Citations 32-35 are irrelevant to the topic of this research.

Answer: As you commented, we remove these references (32-35).

9. TABLES: Satisfactory

Answer: Thank you for your comment.

10. FIGURES: Figure 2 is not clear: a higher quality image is recommended.

Answer: As you recommended, we revised figure 2.

---

## [Editor Report · Decision Letter 1]

15 Jul 2020

Association of adenotonsillectomy with asthma and upper respiratory infection: A Nationwide Cohort Study

PONE-D-20-16108R1

Dear Dr. Kim,

We’re pleased to inform you that your manuscript has been judged scientifically suitable for publication and will be formally accepted for publication once it meets all outstanding technical requirements.

Kind regards,

Giovanni Cammaroto

Academic Editor

PLOS ONE
---

## [Editor Report · Acceptance letter]

17 Jul 2020

PONE-D-20-16108R1 

Association of adenotonsillectomy with asthma and upper respiratory infection: A Nationwide Cohort Study 

Dear Dr. Kim:

I'm pleased to inform you that your manuscript has been deemed suitable for publication in PLOS ONE. Congratulations! Your manuscript is now with our production department. 

Kind regards, 

on behalf of

Dr. Giovanni Cammaroto 

Academic Editor

PLOS ONE